# Clustering Analysis of Multilayer Complex Network of Nanjing Metro Based on Traffic Line and Passenger Flow Big Data

Ming Li [1], Wei Yu [2,*] and Jun Zhang [2]

[1] College of Network and Communication Engineering, Jinling Institute of Technology, Hongjing Road 99#, Nanjing 211169, China; liming@jit.edu.cn
[2] College of Automobile and Traffic Engineering, Nanjing Forestry University, Longpan Road 159#, Nanjing 210037, China; zmars1980@njfu.edu.cn
* Correspondence: yuweicar@163.com

**Abstract:** Complex networks in reality are not just single-layer networks. The connection of nodes in an urban metro network includes two kinds of connections: line and passenger flow. In fact, it is a multilayer network. The line network constructed by the Space L model based on a complex network reflects the geographical proximity of stations, which is an undirected and weightless network. The passenger flow network constructed with smart card big data reflects the passenger flow relationship between stations, which is a directed weighted network. The construction of a line-flow multilayer network can reflect the actual situation of metro traffic passenger flow, and the node clustering coefficient can measure the passenger flow clustering effect of the station on adjacent stations. Combined with the situation of subway lines in Nanjing and card-swiping big data, this research constructs the line network with the Space L model and the passenger flow network with smart card big data, and uses these two networks to construct the multilayer network of line flow. This research improves the calculation method of the clustering coefficient of weighted networks, proposes the concept of node group, distinguishes the inflow and outflow, and successively calculates the clustering coefficient of nodes and the whole network in the multilayer network. The degree of passenger flow activity in the network thermal diagram is used to represent the passenger flow activity of the line-flow network. This method can be used to evaluate the clustering effect of metro stations and identify the business districts in the metro network, so as to improve the level of intelligent transportation management and provide a theoretical basis for transportation construction and business planning.

**Keywords:** transportation; multilayer network; clustering coefficient; complex network; big data

## 1. Introduction

The emergence of the metro has promoted the development of the city and brought convenience to residents' travel. Through the complex network method, taking the metro station as the node and the metro line or passenger flow as the connection between nodes, the line network and passenger flow network can be realized. These two networks are closely linked, which can further form a multilayer line-flow network, reflecting the passenger flow relationship between metro stations in the line network.

Similar to a line-flow network, many complex network models are generally not the single-layer undirected unweighted network, and the connection between nodes has direction and weight, which is composed of multilayer networks that affect each other. For the problem of the multilayer network model, some networks are the analysis target, and some networks are the limiting conditions.

For the multilayer networks composed of different networks, there are many names in related research, such as coupled network, multirelationship network, multilayer network, multiplex network, network of networks, interconnected network, and so on.

The coupled complex network is generally used to describe the synchronization problem in the control system. Li derived the synchronization criterion of two networks with the same topological connectivity [1]. Wu analyzed the problem of synchronous optimization of linear coupled complex networks by selecting appropriate control strategies [2]. Wang proposed a method to identify the parameters and topology of unknown nodes in nonlinear coupled complex networks under random noise disturbance [3].

Multirelationship networks can be used in the analysis of social network. Chen used the multirelationship concept network for searching information [4]. Bin used the multirelationship social network for collaborative recommendation [5]. Some researchers also adopt the concept of network of networks. Li proposed the network of networks to describe the different relationships between real systems [6].

There are many studies on multilayer networks and multiplex networks. De introduced a framework for studying multilayer networks, and discussed several important network parameters and dynamic processes [7]. De studied the empirical interconnection of multilayer networks [8]. De defined the epidemic process on single and multilayer networks, and discussed the main methods for numerical simulation in detail [9].

Mucha developed a general framework of network quality function, which can be used to study the community structure of any multiplex network [10]. Gomez proposed a construction method of matrix, which was helpful to understand the physics of diffusion such as processes on multiplex networks [11].

Some scholars call such networks interconnected networks. Saumell analyzed the spread of epidemics on two interconnected complex networks [12]. Radicchi proved that the process of building independent networks into interconnected networks will undergo a sharp structural change [13].

Some researchers have studied the clustering coefficient of weighted networks, but few have studied the clustering coefficient of directed networks. D. J. Watts proposed a method to calculate the clustering coefficient of undirected unweighted networks [14]. Marc, Onnela, and Holme, respectively, proposed three different methods to calculate the clustering coefficient of undirected weighted networks [15–17]. Guo proposed a method to calculate the clustering coefficient of directed unweighted networks [18].

SaramaKi compared various definitions of clustering coefficients of weighted complex networks and pointed out their advantages and limitations [19]. Zhang compared several calculation methods of clustering coefficient of weighted complex networks, and explained the dependence of local clustering of nodes on node degree and node strength [20]. Wang compared different induction methods of clustering coefficients of weighted undirected graphs [21]. Berenhaut proposed a new method to calculate the clustering coefficients to weighted networks consist of multiple types of nodes [22]. Tabak pointed out that the directed clustering coefficient of complex networks can be used as an indicator of banking system risk [23].

The current research has not unified the appellation of these multilayer networks. There are many kinds of expressions, such as coupled network, multirelationship network, multilayer network, multiplex network, interconnected network, network of networks, and so on. Generally speaking, the two terms of multilayer network and multiplex network are quite common. These studies involve the definition and performance analysis of multilayer networks. However, due to the difference of system dynamics between multilayer networks and single-layer networks, they are usually directed and weighted networks. The original methods and indicators used to analyze complex networks are generally applicable to undirected and unauthorized networks and not to multilayer networks. When it comes to network modeling, due to different specific problems, the multilayer network is also different, so appropriate modeling methods and performance indicators need to be used.

This research involves a multilayer network model in an attempt to calculate the flow-clustering coefficient of a line-flow multilayer network. It is assumed that multilayer networks are composed of different single networks, and the corresponding nodes of these networks are the same, but the connections between nodes are different. The main network

is a globally coupled network, and any two nodes are connected by different flow. The flow can be passenger flow, which is directed and weighted. The secondary network is a line network composed of adjacent nodes.

Nanjing metro has opened a number of metro lines. Nanjing metro connects the urban central area with remote counties, as well as stations, airports, and other transport hubs, providing great convenience for residents to travel, which is suitable for analysis as a typical case. Wei analyzed the performance of Nanjing metro with a supernetwork method [24]. Yu et al. (2019) analyzed the space–time evolution of Nanjing metro network with a complex network [25].

Yu et al. (2019) found that with the evolution of metro network, the robustness of the network is gradually improving [26]. The Nanjing metro management department has also used the automatic ticketing system. Passengers can use the public smart card to take the subway. Wei identified the abnormalities of these swiping card records and proposed the data-filtering process [27]. Wei analyzed passengers' travel preferences and space–time distribution of passenger flow through their swiping card records [28]. Yu classified the factors affecting the passengers' travel time and made the correlation analysis [29].

The Nanjing metro line network and passenger flow network together constitute a line-flow multilayer network. By studying the clustering effect of Nanjing metro stations, we can know the degree of close connection between stations. The metro network can only reflect the clustering of still metro stations. The passenger flow network is a globally coupled network that cannot reflect the true topological structure of metro lines. Only by considering the integration of these two networks can we extract the problem of passenger flow clustering in metro lines, and quantify and evaluate it.

This research describes the clustering problem of nodes in multilayer networks, summarizes the previous studies on the clustering coefficient of complex networks, and proposes the calculation principle and analysis process of the clustering coefficient of a line-flow multilayer network. The clustering coefficient of the node and the multilayer network are used to evaluate the clustering effect of the line-flow network. According to the different flow direction, it can be also divided into inflow and outflow. The research results can be used to analyze the clustering effect of business districts and transportation hubs, improve the level of intelligent transportation management, and provide theoretical support for business and transportation planning.

## 2. Methods

### 2.1. Problem Description

The more metro lines there are, the more obvious the clustering effect of passenger flow is. Using the Space L model of complex network, taking metro stations as nodes and metro lines as the connection between nodes, the complex network of metro lines can be constructed. The Space L model reflects the geographical proximity of metro stations and is an undirected and unweighted network. The metro smart card facilitates passengers' travel and accumulates big data of card swiping, which can record the stations and times of passengers moving on and off. Through smart card big data, we can build a passenger flow network, which is a directed weighted network. Because there is a connection of passenger flow between any two stations, the passenger flow network is also a global network. When the line network and passenger flow network are combined together, it forms a multilayer network of line-flow, and reflects the two connections between line and passenger flow at the same time.

In the performance parameters of complex networks, the clustering coefficient of nodes can be used to represent the clustering effect of nodes, that is, the degree of interconnection between nodes and adjacent nodes. The network clustering coefficient is obtained by averaging the clustering coefficient of nodes. For a single network, the calculation of clustering coefficient is relatively easy. However, for the multilayer network of line flow, determining how to calculate the passenger flow clustering coefficient of nodes in the

metro line network and evaluate the passenger flow clustering effect of subway stations is a problem worthy of attention.

Figure 1 shows a line-flow multilayer network with five nodes, including the passenger flow network and the line network.

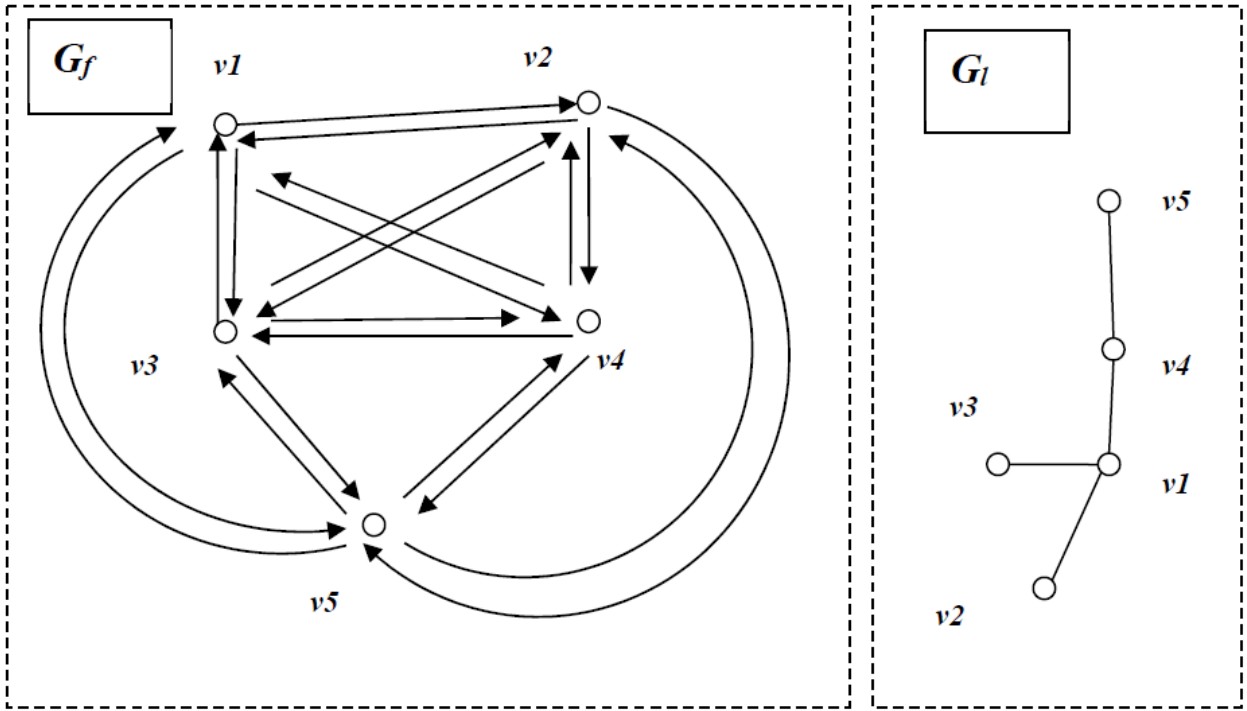

**Figure 1.** Structure of a line-flow multilayer network.

Each network includes five nodes, namely, $v_1$, $v_2$, $v_3$, $v_4$, $v_5$. $G_f$ is the flow network. The connection of any two nodes represents the passenger flow value between nodes. It is a directed weighted network and a globally coupled network. In the line network $G_l$, a node can represent a station, and the connection between any two nodes represents the proximity between nodes. $G_l$ is an undirected unweighted network.

### 2.2. Previous Calculation Methods of Clustering Coefficient

Watts proposed a method to calculate the clustering coefficient of undirected and unweighted networks in Formula (1) [14].

$$C = \frac{1}{N}\sum_{i=1}^{N} C_i = \frac{1}{N}\sum_{i=1}^{N} \frac{2M_i}{k_i(k_i - 1)} \tag{1}$$

In Formula (1), $C_i$ is the clustering coefficient of node $v_i$, and $C$ is the clustering coefficient of the network. It is assumed that $k_i$ nodes are directly connected to nodes $v_i$, so for undirected networks, the maximum number of possible edges among $k_i$ nodes is $[k_i(k_i - 1)]/2$, while the actual number of edges is $M_i$. $N$ is the number of all nodes.

Three different methods of the clustering coefficient of undirected weighted network have been, respectively, proposed [15–17]. These calculation methods comprehensively consider the weight value of the node group composed of the node and its adjacent nodes. Guo proposed a method to calculate the clustering coefficient of directed and unweighted networks in Formulas (2) and (3) [18].

$$C_{in} = \frac{1}{N}\sum C_i^{in} = \frac{1}{N}\sum \frac{M_i^{in}}{[k_i^{in}(k_i^{in} - 1)]} \tag{2}$$

$$C_{out} = \frac{1}{N}\sum C_i^{out} = \frac{1}{N}\sum \frac{M_i^{out}}{[k_i^{out}(k_i^{out} - 1)]} \tag{3}$$

$C_i^{in}$ is the inflow clustering coefficient of node $v_i$, and $C_{in}$ is the inflow clustering coefficient of network. $C_i^{out}$ is the outflow clustering coefficient of node $v_i$, and $C_{out}$ is the outflow clustering coefficient of network. If the inflow degree of a node $v_i$ is $k_i^{in}$, the maximum number of arcs that may exist among $k_i^{in}$ start nodes connected to the node is $k_i^{in}(k_i^{in} - 1)$, but the actual number of arcs is $M_i^{in}$. If the outflow degree of a node $v_i$ is $k_i^{out}$, the maximum number of arcs that may exist among $k_i^{out}$ start nodes connected to the node is $k_i^{out}(k_i^{out} - 1)$, but the actual number of arcs is $M_i^{out}$.

These methods calculate the clustering coefficient by using the complex network method, but they also have some limitations. These methods are not related to the directed weighted network, and the network type is a single network. These methods are used to calculate the clustering coefficient of a line-flow multilayer network.

### 2.3. Calculation Assumption of Clustering Coefficient of Line-Flow Multilayer Network

The calculation assumption of the clustering coefficient of a line-flow multilayer network is as follows.

(1) The clustering coefficient of the node can be divided into inflow and outflow.
(2) The contribution of nodes to the clustering coefficient should be proportional to the weight of the edge.
(3) The network clustering coefficient is the average value of the clustering coefficient of all nodes.
(4) The clustering coefficient of the node or the multilayer network ranges from 0 to 1. The higher the value, the higher the degree of clustering.
(5) When the line network becomes a globally coupled network, the clustering coefficient of the line-flow multilayer network is 1.

### 2.4. Calculation Process of Clustering Effect of Line-Flow Multilayer Network

The clustering effect of the line-flow multilayer network includes the inner and overall flow of node groups, clustering coefficient of node, and the multilayer network. The calculation process is as follows.

(1) Step 1: Establish the flow network according to the directed weighted flow between nodes.

The flow network describes the flow relationship between nodes, and its adjacency matrix is a directed weighted matrix. The corresponding flow adjacency matrix $F = \{f_{ij}\}_{N \times N}$ can be defined as Formula (4).

$$f_{ij} = w_{ij} \tag{4}$$

where $w_{ij}$ is the weight of the edge $e_{ij}$, representing the flow from node $i$ to node $j$.

(2) Step 2: Establish the line network according to the adjacent stations in the line network. It is the undirected unweighted network.

Figure 2 shows the Space L model established by metro lines. The Space L model based on a complex network is applicable for modeling the line network [25]. If any two nodes are adjacent to each other on a certain line, the two nodes establish the proximity relationship.

(3) Step 3: Calculate the total flow of the node groups.

The concept of a node group is defined as the node and its adjacent nodes in Figure 3. The total flow of the node group reflects the interaction between the node group and the whole network.

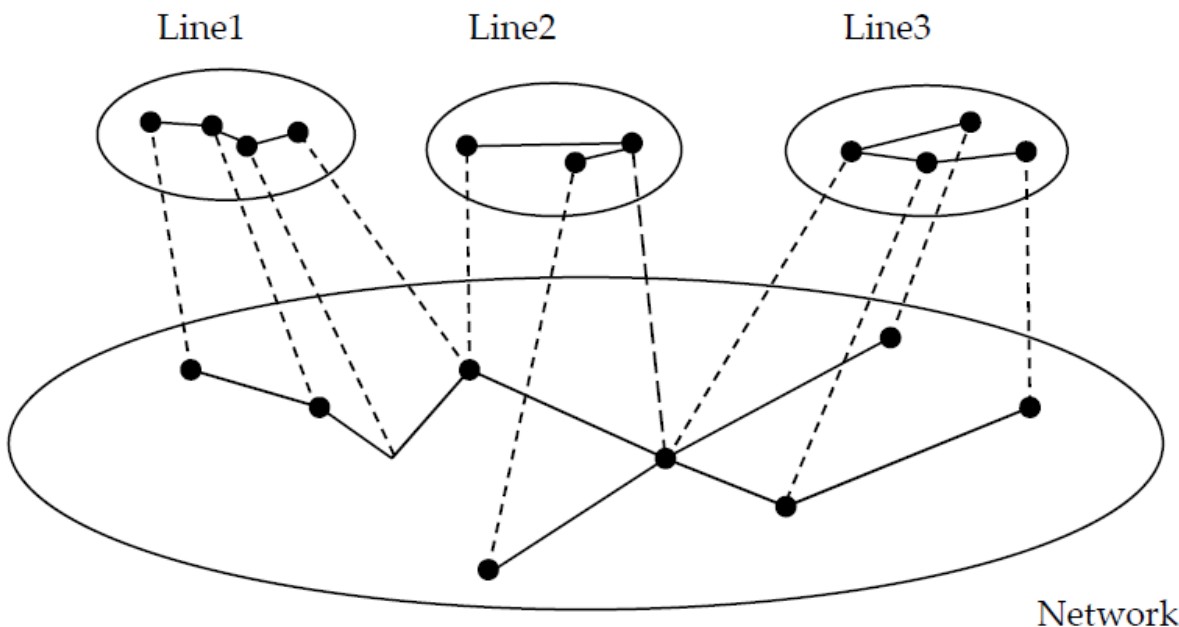

**Figure 2.** Space L model established by metro lines.

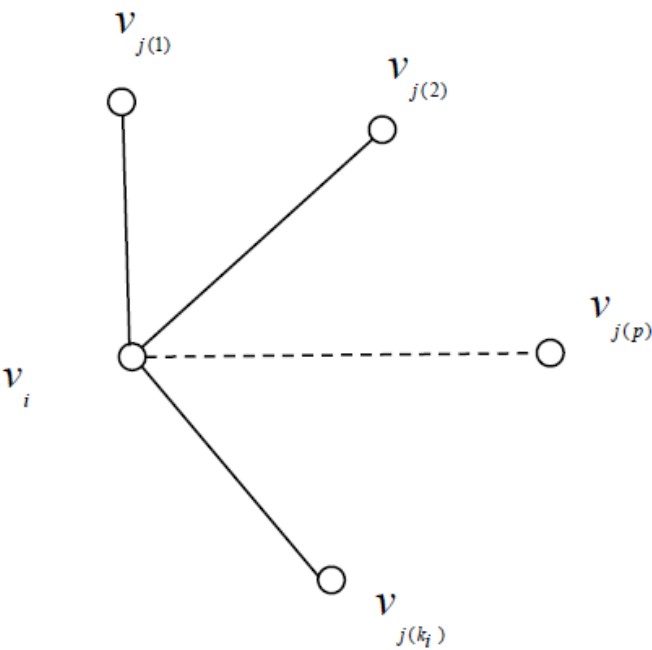

**Figure 3.** Definition of the node group.

Suppose that in the line network $G_l$, the node degree of node $v_i$ is $k_i$, which means that the node $v_i$ has connected $k_i$ nodes, and the code of other nodes connected is $j(p)$. In the adjacent matrix $L = \{l_{ij}\}_{N \times N}$, $l_{i\ j(p)} = 1$, $p = 1, 2, \cdots, k_i$.

Figure 4 shows the inner flow and total flow of the node group, as in Figure 1. In the line network $G_t$, the node degree of node $v_1$ is 3, which means that the node $v_1$ has three connected nodes, namely, $v_2$, $v_3$, and $v_4$. The node $v_1$ group can be defined as $v_1^{group} = (v_1, v_2, v_3, v_4)$. In the flow network $G_f$, the solid line represents the inner flow of the node $v_1$ group, the dotted line represents the external flow of the node $v_1$ group, and the sum of the two represents the total flow of the node $v_1$ group. In the same way, we can obtain the inner flow and total flow of other node groups.

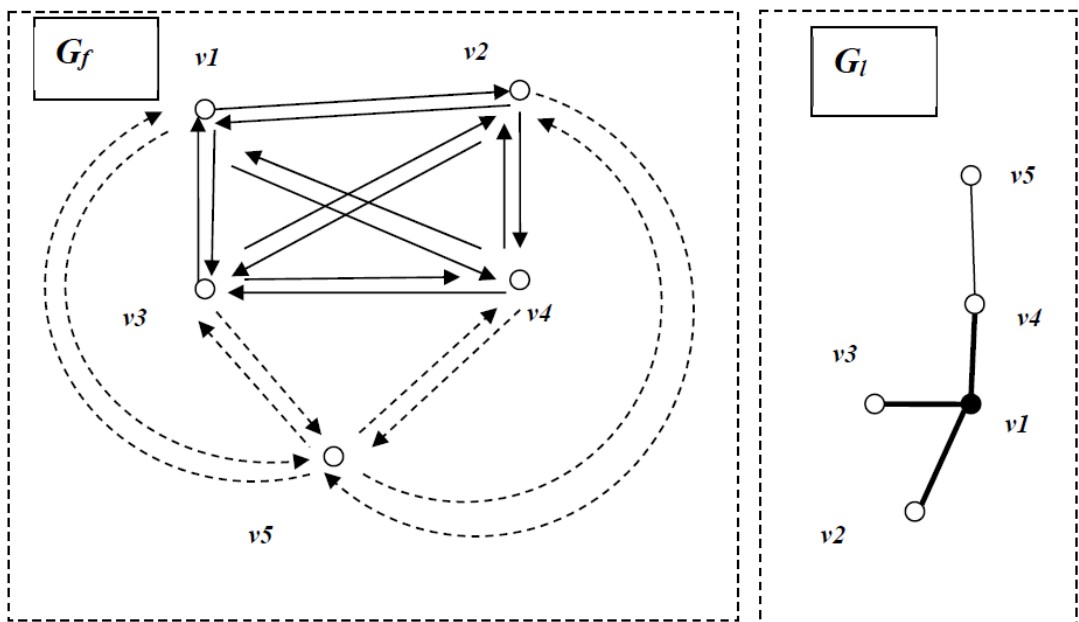

**Figure 4.** Inflow and total flow of the node group.

In the flow network $G_f$, we define the inflow from node $i$ to node $j$ as $f_{ij}^{in}$, and outflow as $f_{ij}^{out}$, then $f_{ij}^{in} = w_{ji}$, and $f_{ij}^{out} = w_{ij}$.

The flow between the same nodes is zero. Formulas (5)–(8) show the total flow calculation method of the node group. Formula (4) shows the calculation of total inflow of the node $i$.

$$f_{node\ total(i)}^{in} = \sum_{j=1}^{N} f_{ij}^{in} \tag{5}$$

Formula (5) shows the calculation of total inflow of other nodes connected with the node $i$.

$$f_{other\ node\ total(i)}^{in} = \sum_{p=j(1)}^{j(k_i)} f_{node\ total(p)}^{in} = \sum_{p=j(1)}^{j(k_i)} \sum_{j=1}^{N} f_{pj}^{in} \tag{6}$$

Formula (6) shows the calculation of total inflow of the node $i$ group.

$$f_{group\ total(i)}^{in} = f_{node\ total(i)}^{in} + f_{other\ node\ total(i)}^{in} = \sum_{j=1}^{N} f_{ij}^{in} + \sum_{p=j(1)}^{j(k_i)} \sum_{j=1}^{N} f_{pj}^{in} \tag{7}$$

In the same way, Formula (7) shows the calculation of total outflow of the node $i$ group.

$$f_{group\ total(i)}^{out} = f_{node\ total(i)}^{out} + f_{other\ node\ total(i)}^{out} = \sum_{j=1}^{N} f_{ij}^{out} + \sum_{p=j(1)}^{j(k_i)} \sum_{j=1}^{N} f_{pj}^{out} \tag{8}$$

The flow network is a globally coupled network, so the whole inflow and the whole outflow of the flow network are equal.

(4)   Step 4: Calculate the inner flow of the node groups, including inner inflow and inner outflow. The flow between the same nodes is zero.

The inner flow of the node group reflects the interaction of the nodes in the node group. Formulas (9)–(11) show the inner flow calculation method of the node group. Formula 8 shows the calculation of inner inflow of the node *i*.

$$f_{node\ inner(i)}^{in} = \sum_{j=j(1)}^{j(k_i)} f_{ij}^{in} \tag{9}$$

Formula (9) shows the calculation of inner inflow of other nodes connected with the node *i*.

$$f_{other\ node\ inner(i)}^{in} = \sum_{p=j(1)}^{j(k_i)} f_{node\ inner(p)}^{in} = \sum_{p=j(1)}^{j(k_i)} \sum_{j=j(1)}^{j(k_i)} f_{pj}^{in} + \sum_{p=j(1)}^{j(k_i)} f_{pi}^{in} \tag{10}$$

Formula (10) shows the calculation of inner inflow of the node *i* group.

$$f_{group\ inner(i)}^{in} = f_{node\ inner(i)}^{in} + f_{other\ node\ inner(i)}^{in} = \sum_{j=j(1)}^{j(k_i)} f_{ij}^{in} + \sum_{i=j(1)}^{j(k_i)} \sum_{j=j(1)}^{j(k_i)} f_{ij}^{in} + \sum_{p=j(1)}^{j(k_i)} f_{pi}^{in} \tag{11}$$

The inner flow network of each node group is a small globally coupled network, so the inner outflow of the node *i* group is equal to the inner outflow of the node *i* group.

Formula (12) shows the calculation of inner outflow of the node *i* group.

$$f_{group\ inner(i)}^{out} = f_{group\ inner(i)}^{in} \tag{12}$$

(5)　Step 5: Calculate the flow clustering coefficient of the node, including inflow clustering coefficient and outflow clustering coefficient.

The clustering coefficient of the node is the ratio of inner flow and total flow of node group.

The inflow clustering coefficient of the node *i* group is the ratio of the inner inflow to the total inflow of the node *i* group, as shown in Formula (13).

$$C_{node(i)}^{in} = \frac{f_{group\ inner(i)}^{in}}{f_{group\ total(i)}^{in}} \tag{13}$$

The outflow clustering coefficient of the node *i* group is the ratio of the inner outflow to the total outflow of the node *i* group, as shown in Formula (14).

$$C_{node(i)}^{out} = \frac{f_{group\ inner(i)}^{out}}{f_{group\ total(i)}^{out}} \tag{14}$$

(6)　Step 6: Calculate the flow clustering coefficient of the line-flow multilayer network, including inflow clustering coefficient and outflow clustering coefficient.

The inflow clustering coefficient of the line-flow multilayer network is the average value of inflow clustering coefficient of all nodes, as shown in Formula (15).

$$C^{in} = \frac{1}{N} \sum_{i=1}^{N} C_{group(i)}^{in} \tag{15}$$

The outflow clustering coefficient of the line-flow multilayer network is the average value of outflow clustering coefficient of all nodes, as shown in Formula (16).

$$C^{out} = \frac{1}{N} \sum_{i=1}^{N} C_{group(i)}^{out} \tag{16}$$

## 3. Results

### 3.1. Situation of Nanjing Metro

Urban development will change the way that residents travel. Nanjing has formed a perfect three-dimensional transportation system. The information and communication technology has also changed the travel mode of residents [30]. The different structures of an urban road network have a deep impact on the mode of transportation [31]. Many residents choose to travel by metro [32]. For health reasons, some residents may choose to walk [33]. E-commerce also affects the logistics of the last kilometer [34]. Residents can choose a variety of ways to obtain express delivery [35].

The case of this research is the line-flow multilayer network of Nanjing metro. Nanjing metro had opened 7 lines with 128 stations by 2017. Table 1 shows the situation of Nanjing metro lines, including line name, opening time, number of stations, and length.

**Table 1.** Situation of Nanjing metro lines.

| Opening Sequence | Number of Stations | Length (km) | Opening Year |
|---|---|---|---|
| 1 | 27 | 38.9 | 2005 |
| 2 | 26 | 37.95 | 2010 |
| 10 | 14 | 21.6 | 2014 |
| S1 | 8 | 37.3 | 2014 |
| S8 | 17 | 45.2 | 2014 |
| 3 | 29 | 44.9 | 2015 |
| 4 | 18 | 33.8 | 2017 |

Figure 5 shows the Nanjing metro network map in 2017, with different colors to represent different lines. Each station is marked with a corresponding number, representing the number in the Nanjing metro management system. These codes are closely related to the opening sequence of metro stations. The Nanjing metro network has an obvious central effect, among which the area formed by the intersection of Lines 1, 2, 3, and 4 is the core area. The area formed by the intersection of Lines 1 and 3 is the central area, which is much larger than the core area. Other lines radiate from the central area to the surrounding areas, including some emerging areas and suburb counties.

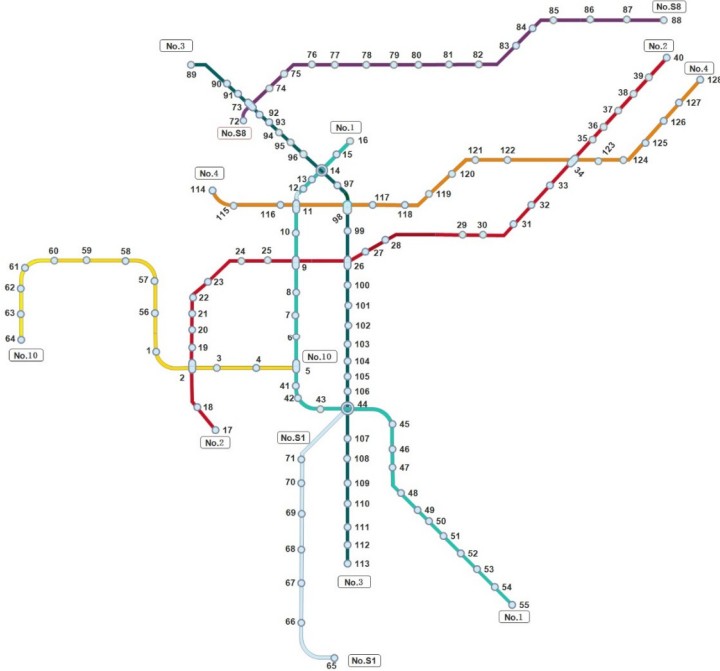

**Figure 5.** Nanjing metro network map.

### 3.2. Flow Network of Nanjing Metro

In the flow network, the node represents the metro station, and the connection between nodes is expressed by the passenger flow. The network modeling is based on the traffic smart card data on 13 February 2017, with a total of more than one million. The smart card data records the OD (origination–destination) travel of the passengers, including card number, card type, inbound and outbound stations, inbound and outbound time, etc. The data should be filtered before the analysis of the clustering coefficient. The passenger flow network is a directed weighted network and a globally coupled complex network.

Figure 6 shows the passenger flow distribution between Nanjing metro stations on 13 February 2017. After data filtering, there are 1,218,423 effective swiping card records in Nanjing metro network with 128 stations. The maximum flow between stations exceeds 8000 persons. The minimum flow is zero, which means that there is no passenger flow between some stations. Most of the passenger flow is within 100 persons, and some are 200–500 persons.

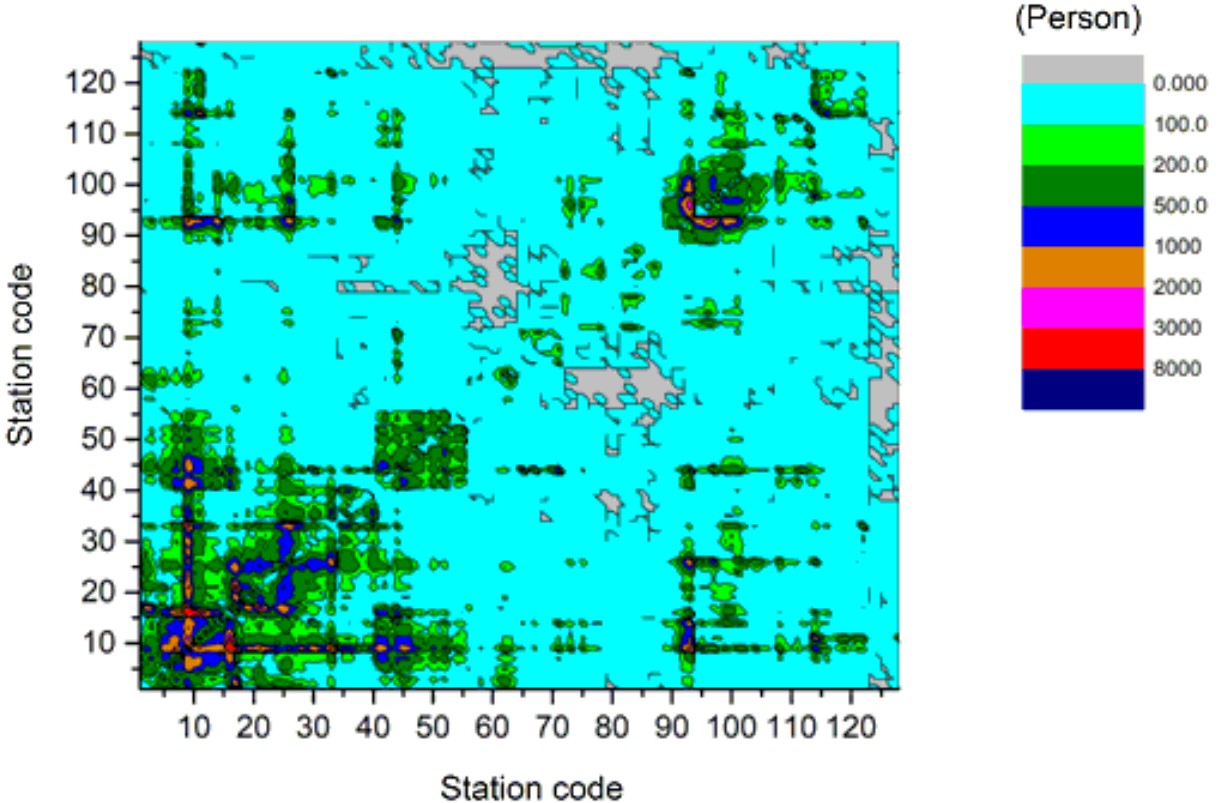

**Figure 6.** Passenger flow distribution between Nanjing metro stations.

### 3.3. Line Network of Nanjing Metro

Figure 7 shows the Space L model of the Nanjing metro network. When the Space L model is established with the complex network method, an edge between the nodes can be built through the two adjacent stations on a line, so as to obtain the topology map of the whole subway network, which reflects the geographical proximity of the stations. The research used Ucinet software to establish the network matrix, and used Netdraw to draw the corresponding figure; each station has different numbers on the graphics. The stations and their codes in Figures 5 and 7 are closely related.

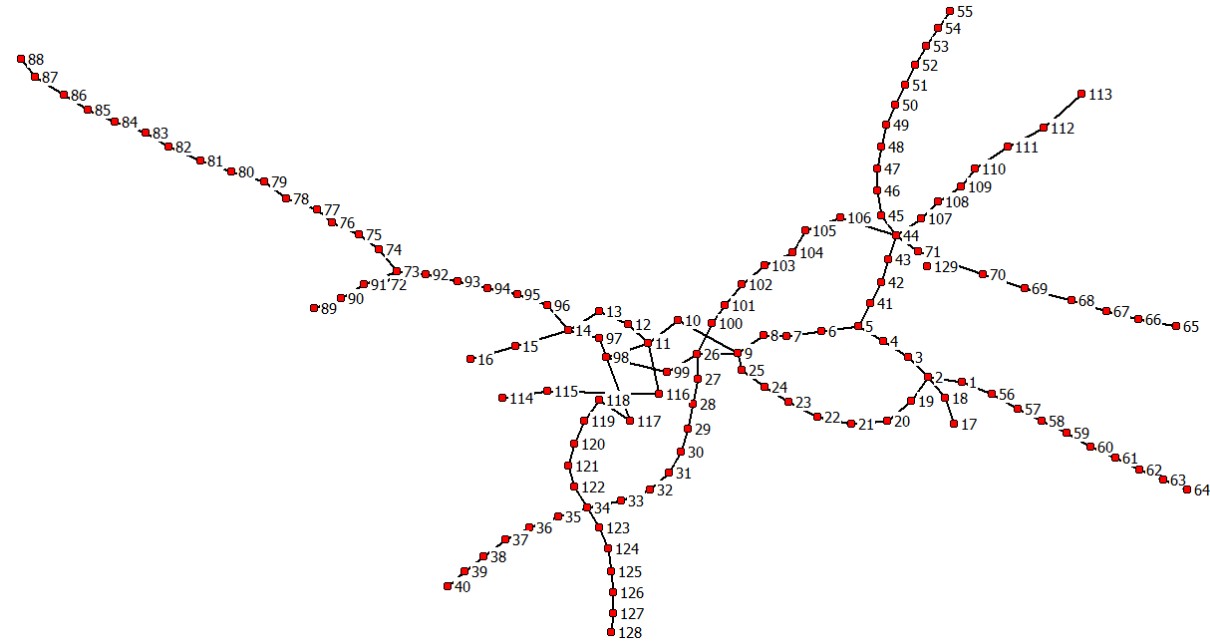

**Figure 7.** Space L model of Nanjing metro network.

Figure 8 shows the complex network model of Nanjing metro node groups. Each station and its adjacent stations form a node group. All nodes in each node group are interconnected, thus forming an inner network and a small globally coupled network.

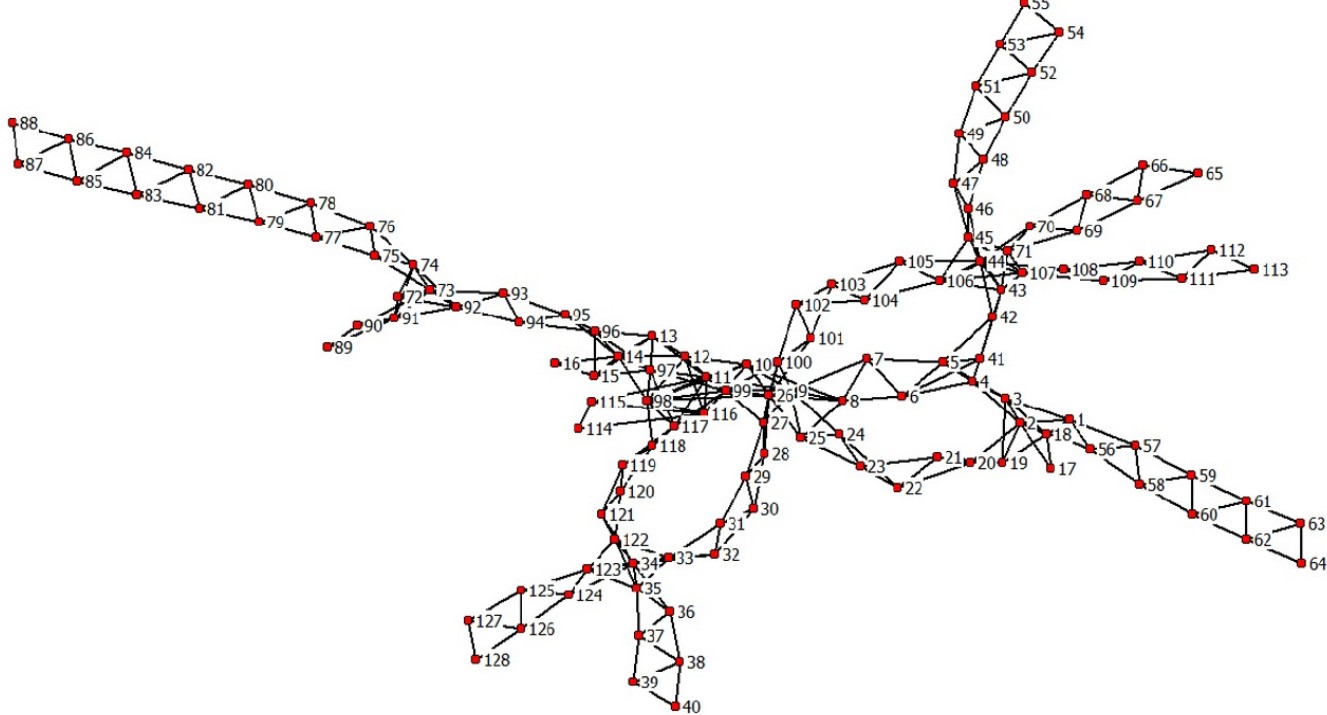

**Figure 8.** Complex network model of the inner relationship of Nanjing metro node groups.

*3.4. Inflow and Outflow Clustering Coefficient of the Stations in Line-Flow Multilayer Network*

Figure 9 shows the distribution of the inflow and outflow clustering coefficients of Nanjing metro nodes, and the trends of the two are basically the same. The clustering coefficient of the node is the ratio of inner flow to total flow of the node group. Because the line network of Nanjing metro has not been completed and the network density is not large,

the clustering coefficients of the node groups are not large, and the maximum value is close to 0.125. The clustering coefficient of most node groups is less than 0.050. The distribution of clustering coefficients has no obvious central effect, and it is not concentrated in the core area and main lines.

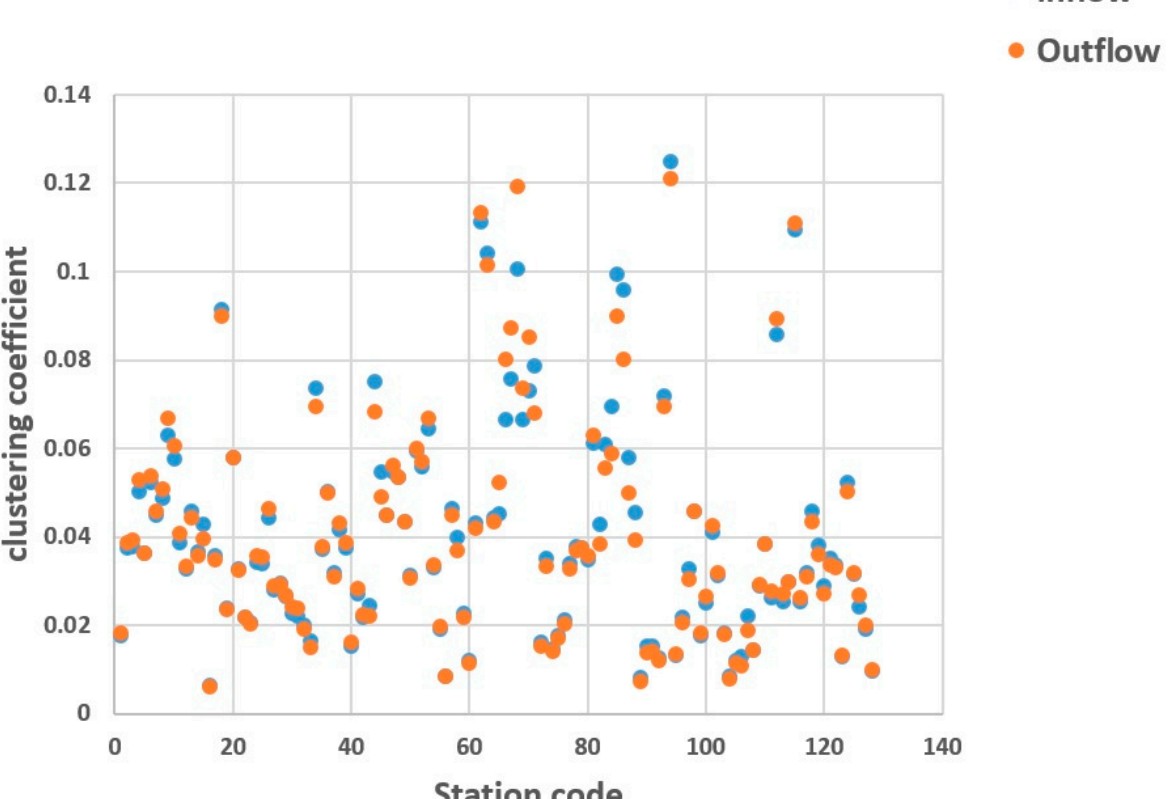

**Figure 9.** Distribution of inflow and outflow clustering coefficients of Nanjing metro nodes.

Figure 10 shows the thermal diagram of inflow clustering coefficient of Nanjing metro nodes. In order to express the difference of the clustering coefficient, the line color in Figure 5 is set to gray, and the five colors of green, blue, yellow, orange, and red are used to represent different intervals of clustering coefficient. The green clustering coefficient interval is (0, 0.025], blue clustering coefficient interval is (0.025, 0.050], yellow clustering coefficient interval is (0.050, 0.075], orange clustering coefficient interval is (0.075, 0.100], and red clustering coefficient interval is greater than 0.100.

Red stations are 62, 63, 68, 94, and 115. Orange stations are 18, 66, 67, 70, 85, 86, and 112. These two kinds of nodes are basically distributed in the extension line or the end of the line of Nanjing metro. This shows that the public transportation around these nodes is inconvenient and far away, and passengers tend to travel to the nearby stations through subway lines, so the clustering coefficient is high. Yellow stations are generally distributed in line transfer stations or intermediate stations. Blue stations are the most widely distributed. Green stations are concentrated around 73 and 44; the former is the intersection area of Line 3 and Line S8, and the latter is near the new railway station in Nanjing. The subway stations in these two parts are relatively short, the public transportation is also convenient, and the clustering effect is not obvious.

Figure 11 shows the thermal diagram of outflow clustering coefficient of Nanjing metro nodes. The color setting of the station is the same as Figure 10. Red stations are 62, 63, 68, 94, and 115. Orange stations are 18, 67, 71, 85, 86, and 112. Yellow stations are generally distributed in line transfer stations or intermediate stations. Blue stations are the most widely distributed. Green stations are concentrated around 73 and 44. The

distribution diagram of outflow clustering coefficient is basically the same as that of inflow clustering coefficient.

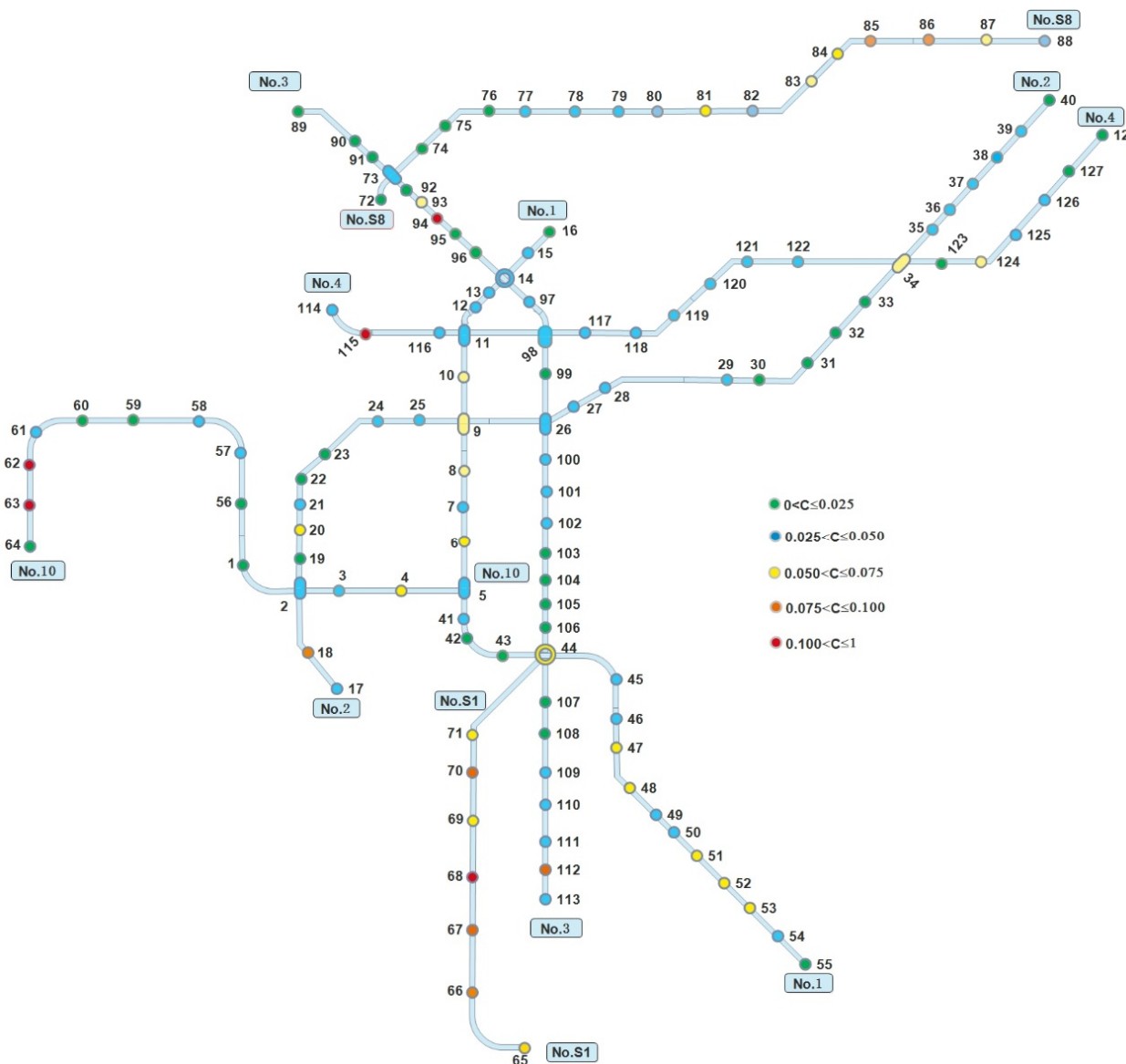

**Figure 10.** Thermal diagram of inflow clustering coefficient of Nanjing metro nodes.

### 3.5. Inflow and Outflow Clustering Coefficient of Line-Flow Multilayer Network

Table 2 shows the clustering coefficient of the line-flow multilayer network of Nanjing metro in five working days. The clustering coefficient of the line-flow network is the average of the clustering coefficients of all nodes. The period is from 13 February to 17 February 2017. After data filtering, there are about 1.2 million swiping card records in the Nanjing metro network every day. The inflow and outflow of the whole subway network are equal. The inflow clustering coefficient of line-flow network is 0.0405–0.0426, and the outflow clustering coefficient of line-flow network is 0.0404–0.0427. The values of the inflow and outflow clustering coefficients of the same day are basically the same.

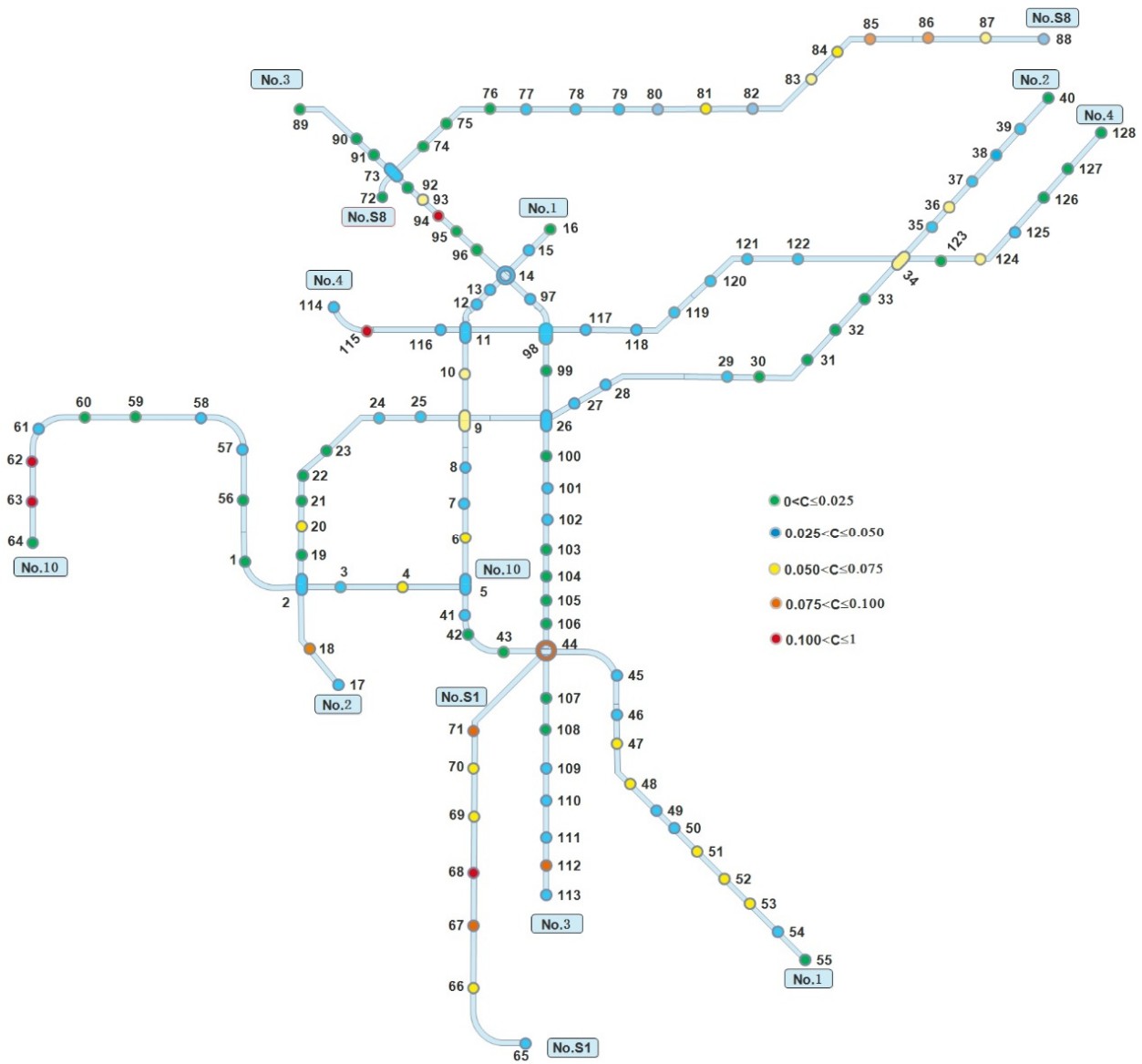

**Figure 11.** Thermal diagram of outflow clustering coefficient of Nanjing metro nodes.

**Table 2.** Network clustering coefficient of line-flow multilayer network of Nanjing metro in five working days.

| Date | Week | Number after Filtering | Network Inflow Clustering Coefficient | Network Outflow Clustering Coefficient |
|---|---|---|---|---|
| 2.13 | Monday | 1,218,423 | 0.0405 | 0.0404 |
| 2.14 | Tuesday | 1,294,948 | 0.0426 | 0.0427 |
| 2.15 | Wednesday | 1,229,704 | 0.0412 | 0.0414 |
| 2.16 | Thursday | 1,192,083 | 0.0409 | 0.0409 |
| 2.17 | Friday | 1,313,340 | 0.0408 | 0.0414 |

The reason is that there is a certain regularity in the travel of passengers in Nanjing metro, and they usually travel back and forth between fixed working places and residences, so as to achieve the flow balance between early peak and late peak between stations. In terms of the whole day, the inflow and outflow between stations are basically the same, which also leads to the equality of the inflow and outflow clustering coefficients. The small

value of clustering coefficient is due to the fact that the Nanjing metro network is still developing, and the network density is not enough. When the line network becomes a globally coupled network, the clustering coefficient of the line-flow multilayer network can reach 1.

## 4. Discussion

The metro line network can be established using the Space L model of complex networks. The passenger flow network can be established using the big data of smart cards. Line network and passenger flow network constitute a multilayer network of line flow. In this multilayer network, determining how to evaluate the passenger flow clustering effect of nodes in the line-flow network is the problem that this research attempts to solve. The clustering coefficient of complex networks is an effective index to evaluate the clustering effect of nodes, which reflects the close relationship between nodes and adjacent nodes.

The previous research on clustering coefficient is basically directed at single-layer networks, including undirected and unweighted networks, undirected weighted networks, directed and unweighted networks, and so on. The passenger flow network is a directed weighted network, and the line network is an undirected and unweighted network. The calculation method of the clustering coefficient of this multilayer network is basically not seen.

This research proposes the analysis process and calculation method of the clustering coefficient of a multilayer network, makes a case analysis combined with the big data of Nanjing metro line and smart card, and shows the passenger flow clustering effect of Nanjing metro nodes with thermal diagram. The clustering coefficient is divided into inflow coefficient and outflow coefficient. The results show that the two coefficients of Nanjing metro nodes are similar, and the degree of clustering is high in some nodes. This shows that the passenger flow interaction between these nodes and adjacent nodes is relatively frequent.

In addition, the clustering coefficient of the line-flow multilayer network can be obtained by the average value of the clustering coefficient of all nodes. This research traces the clustering coefficients of the Nanjing subway network of five days in a week. These coefficients basically fluctuate in a small range, which shows that metro passengers have a stable travel rule. Identifying the clustering of a line-flow network can identify the traffic circle and business circle of the city, so as to carry out the geographical planning of the city.

## 5. Conclusions

This method for calculating the clustering coefficient of multilayer complex networks can also be used for other multilayer networks with similar structure. The flow in the flow network can be traffic flow, material flow, or trade flow. The connected relationship in the line network can be a transportation line, a supply chain, or a trade alliance. The clustering effect of these multilayer networks can reflect the impact of geographical or commercial relations on various flow.

The metro line network reflects the geographical proximity of nodes. The passenger flow network reflects the exchange relationship of passenger flow between nodes. By integrating geographic networks and traffic networks, a dual-layer network of line flow is formed. When calculating the clustering coefficient of the multilayer network, firstly, a node community is formed by a node and its geographically adjacent points, and then the passenger flow of the community in the whole network is divided by the passenger flow of the nodes in the community, so as to obtain the clustering coefficient of the node. This parameter actually reflects the siphon effect of the node on adjacent nodes. The greater the clustering coefficient of the node, the stronger the siphon effect, and the greater the attraction of this node to the surrounding nodes.

In economics, the siphon effect reflects the transfer of production factors from small- and medium-sized cities to central cities in regional economic development due to the development gradient difference between cities. Using this concept, in transportation

networks, the siphon effect reflects the effect of passenger flow shifting from secondary nodes to central nodes due to different traffic conditions, location, and other factors at metro stations.

The transportation network can be a metro network, public bus network, high-speed rail network, highway network, etc. These networks can further form a weighted three-dimensional network. The flow in a transportation network can be either passenger flow or cargo flow. This method can be used to analyze the clustering effect of multilayer networks such as transportation flow.

This method can further analyze the siphon effect of multilayer networks composed of relationship networks and flow networks. Nodes can be transportation nodes, individuals, companies, cities, or countries. Relationship networks can be geographic networks, social networks, collaboration networks, or trade networks. Flow networks can include product flow, economic flow, social flow, etc. Due to the fact that flow is divided into inflow and outflow, this method can also effectively evaluate the directionality of the siphon effect.

**Author Contributions:** M.L. designed the research and wrote the paper. W.Y. performed the data collection and analysis. J.Z. edited and modified the paper. All authors have read and agreed to the published version of the manuscript.

**Funding:** This research was funded by National Natural Science Foundation of China, grant number 71701099.

**Institutional Review Board Statement:** Not applicable.

**Informed Consent Statement:** Written informed consent has been obtained from the patient(s) to publish this paper.

**Data Availability Statement:** No additional data are available.

**Conflicts of Interest:** The authors declare that they have no competing interests.

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
