# Peer review of "Clustering Analysis of Multilayer Complex Network of Nanjing Metro Based on Traffic Line and Passenger Flow Big Data"

_sustainability, doi:10.3390/su15129409_

Round 1

Reviewer 1 Report

1. no clear problem statements, should add.

2. add a recommendation to the work

3. the values of the clustering coefficient for Friday are different as compared to other days between inflow and outflow, is there any reason for that?

4. is there any range for clustering coefficient based on previous research?

5. What is the main contribution added by the research? 6. Did the obtained results or data support the conclusions? 7. Does the title reflect precisely the subject area or problem to consider? 8. Is the number of papers in research appropriate? 9. highlight the gaps in the current understanding of the problem and knowledge.

need to consider some grammar mistakes in the paper 

Author Response

  1. No clear problem statements, should add.

Answer:The paper summarizes and supplements the problem statement.

  1. add a recommendation to the work

Answer:The recommendation has been added.

  1. the values of the clustering coefficient for Friday are different as compared to other days between inflow and outflow, is there any reason for that?

Answer:Table 2 shows the statistical data for five days. Except for Thursday, the clustering coefficients of inflow and outflow on other days are slightly different. This is because the majority of passengers traveling every day have fixed commuting habits, resulting in similar clustering coefficients but with small differences.

  1. Is there any range for clustering coefficient based on previous research?

Answer:According to complex network theory, the numerical range of clustering coefficient is between 0 and 1.

  1. What is the main contribution added by the research?

Answer:The main contributions of the study were supplemented in the discussion.

  1. Did the obtained results or data support the conclusions?

Answer:The results and data obtained show that there is a certain degree of clustering of passenger flow in the subway network. These passenger flows are generally concentrated at more active subway stations.

  1. Does the title reflect precisely the subject area or problem to consider?

Answer:The title refers to other research papers and basically reflects keywords such as passenger flow, subway network, and multi-layer network.

  1. Is the number of papers in research appropriate?

Answer:The reference papers basically reflect the relevant research of this paper.

  1. highlight the gaps in the current understanding of the problem and knowledge.

Answer:This section is supplemented during the discussion.

Reviewer 2 Report

The paper investigates complex transport metro networks, namely, the connections of lines and passenger flows of an urban metro network in Nanjing. The line network is constructed by the so-called “Space L model” and the flows are reconstructed from users’ card/ticket validation data, both facilitating the calculation of the clustering coefficient of nodes.

The title is clear and comprehensive. The state of the art is well presented and summarized; however, it seems a bit outdated. I would suggest adding the most recent advances in this regard. (for instance, in terms of the general framework of network quality function, and the assessment of it through different indicators, several previous papers are relevant and could be added to this section (e.g., see https://doi.org/10.1016/j.compenvurbsys.2022.101805, also https://doi.org/10.3390/su14020622.

Also, more recent papers directly related to complex networks could be added.

Methods are concise and clear, however, there are several sentences/parts, where readability could be improved, such as for instance:

Three methods are respectively put forward the calculation method of the clustering coefficient of undirected weighted network[15-17];

However, for the multilayer network of line-flow, how to calculate the passenger flow clustering coefficient of nodes in the metro line network and evaluate the passenger flow clustering effect of subway stations is a problem worthy of attention.

I would suggest a revision of this section to avoid too complex sentences

Results: figure 6 is too small; the legend of the chart should include the full unit – persons per day (person(s)/day)

Conclusion/Discussion: there is no discussion about the applicability of the method and the possible impacts of its use.  I suggest adding a section to explain the practical aspects of the results/used method and the  domains that can benefit from that

Author Response

1、Methods are concise and clear, however, there are several sentences/parts, where readability could be improved, such as for instance:

Three methods are respectively put forward the calculation method of the clustering coefficient of undirected weighted network[15-17];

However, for the multilayer network of line-flow, how to calculate the passenger flow clustering coefficient of nodes in the metro line network and evaluate the passenger flow clustering effect of subway stations is a problem worthy of attention.

I would suggest a revision of this section to avoid too complex sentences

Answer:The content of this section has been modified.

2、Results: figure 6 is too small; the legend of the chart should include the full unit – persons per day (person(s)/day)

Answer:The number of people represented by colors on the heat map is shown in the upper right corner of Figure 6.

3、Conclusion/Discussion: there is no discussion about the applicability of the method and the possible impacts of its use.  I suggest adding a section to explain the practical aspects of the results/used method and the  domains that can benefit.

Answer:The paper has added a section on discussion.

Reviewer 3 Report

The study is interesting and effectively contributes to our understanding of pedestrian flow and network analysis within the domain of urban transport systems. I have a minor suggestion for consideration which, I believe, will further enhance the clarity of the methodology section:
- It would be beneficial to arrange the presented equations (formulas) and their respective descriptions into a tabular format. This will augment the readability of this section and allow for more effortless referencing of the methods used.

Author Response

1、It would be beneficial to arrange the presented equations (formulas) and their respective descriptions into a tabular format. This will augment the readability of this section and allow for more effortless referencing of the methods used.

Answer:I have carefully considered your suggestion. But the formulas in this paper are generally relatively long and have many parameters. If these formulas and their explanations are arranged in a table, it may not look suitable.

Round 2

Reviewer 2 Report

The discussion section is usually put before the conclusions. The discussion as set now in the revised text is not really making things any clearer in terms of the impact and the actual added value of the method developed.

I am leaving the final decision to the editors.

Author Response

The discussion section is usually put before the conclusions. The discussion as set now in the revised text is not really making things any clearer in terms of the impact and the actual added value of the method developed.

Answer:I have adjusted the order of the conclusion and discussion, and the corresponding content has also been adjusted.